Variation in phenology of hibernation and reproduction in the endangered New Mexico meadow jumping mouse (Zapus hudsonius luteus)

Frey Jennifer K. jfrey@nmsu.edu
Department of Fish, Wildlife, and Conservation Ecology, New Mexico State University , Las Cruces, NM , United States of America
Frey Biological Research , Radium Springs, NM , United States of America
Kramer Donald
Electronic publication date: 2015 Aug 6
Publication date: 2015
Volume: 3
Electronic Location ID: e1138
Received 2015 May 6; Accepted 2015 Jul 9
Copyright: © 2015 Frey
Copyright year: 2015
Copyright holder: Frey
License: This is an open access article distributed under the terms of the Creative Commons Attribution License, which permits unrestricted use, distribution, reproduction and adaptation in any medium and for any purpose provided that it is properly attributed. For attribution, the original author(s), title, publication source (PeerJ) and either DOI or URL of the article must be cited.
License URL: https://creativecommons.org/licenses/by/4.0/

Keywords: Life cycle, New Mexico meadow jumping mouse, Zapus hudsonius luteus, Hibernation, Activity season, Reproduction, Endangerd species, Geographic variation, Elevation, Phenology

Funding: This research was not directly supported by a funding source

==============================
Hibernation is a key life history feature that can impact many other crucial aspects of a species’ biology, such as its survival and reproduction. I examined the timing of hibernation and reproduction in the federally endangered New Mexico meadow jumping mouse (Zapus hudsonius luteus), which occurs across a broad range of latitudes and elevations in the American Southwest. Data from museum specimens and field studies supported predictions for later emergence and shorter active intervals in montane populations relative to lower elevation valley populations. A low-elevation population located at Bosque del Apache National Wildlife Refuge (BANWR) in the Rio Grande valley was most similar to other subspecies of Z. hudsonius: the first emergence date was in mid-May and there was an active interval of 162 days. In montane populations of Z. h. luteus, the date of first emergence was delayed until mid-June and the active interval was reduced to ca 124–135 days, similar to some populations of the western jumping mouse (Z. princeps). Last date of immergence into hibernation occurred at about the same time in all populations (mid to late October). In montane populations pregnant females are known from July to late August and evidence suggests that they have a single litter per year. At BANWR two peaks in reproduction were expected based on similarity of active season to Z. h. preblei. However, only one peak was clearly evident, possibly due to later first reproduction and possible torpor during late summer. At BANWR pregnant females are known from June and July. Due to the short activity season and geographic variation in phenology of key life history events of Z. h. luteus, recommendations are made for the appropriate timing for surveys for this endangered species.

Introduction

Hibernation is an adaptive strategy that some mammals use to cope with long-term seasonal limitations in food or water (Davis, 1976; Heldmaier, Ortmann & Elvert, 2004; Ruf & Geiser, 2014). Hibernation is characterized by prolonged multiday bouts of torpor during which the animal’s metabolic rate is significantly lowered, body temperature decreases toward ambient temperature, and many physiological functions cease (Williams et al., 2011; Ruf & Geiser, 2014). Thus, while hibernation can confer profound savings of water and energy, and it has other indirect benefits such as increased survival through predator avoidance, it also bears costs related to altered physiological functions and reduced opportunity for reproduction (Ruf & Geiser, 2014). Despite a great deal of research on the physiology of hibernation, there has been disproportionately little research on the ecological consequences of hibernation (Lane, 2012). Elucidating the timing and duration of biological events associated with hibernation is central to understanding its ecological consequences, because hibernation is tightly linked to other crucial aspects of a species’ biology (Kirkland & Kirkland, 1979; Turbill, Bieber & Ruf, 2011). Recovery from hibernation, reproduction, and sequestering energy reserves to enter and survive the next hibernation all must happen within a brief active period during the warmer months. Thus, timing of emergence from hibernation influences subsequent timing of reproduction, number of litters possible, timing of entrance into hibernation (i.e., immergence), and ultimately overwinter survivorship (e.g., Muchlinski, 1988; Dobson, Badry & Geddes, 1992; Ozgul et al., 2010; Sheriff et al., 2011). Thus, understanding the phenology of hibernation and reproduction is central to understanding the life history of hibernating species. Such questions have gained increased importance due to the potential for altered phenology and mismatches in the phenology of interacting species as a consequence of climate change (e.g., Inouye et al., 2000; Lane, 2012; Boutin & Lane, 2013; Sheriff et al., 2011; Sheriff et al., 2013).

Most studies on variation in phenology of the annual cycle of mammalian hibernators have been conducted on ground squirrels (Sciuridae: Marmotini) and dormice (Gliridae). The jumping mice (Dipodidae: Zapodinae) are another group of small mammal hibernators that are confronted with strongly seasonal environments in the northern temperate zone. However, there has been comparatively little research on hibernation in jumping mice, especially with respect to variation in phenology of the annual cycle (e.g., Lyman, Willis & Malan, 1982). In comparison with most ground squirrels and dormice, jumping mice have much smaller body size (<25 g) and they have not been reported to cache food (Vander Wall, 1990). Although terrestrial mammal hibernators usually are small, in part to facilitate use of underground burrows that are buffered from thermal extremes, hibernators benefit from relatively large body size (while still remaining small enough to burrow) because the principle source of energy for most hibernators during torpor is body fat (Humphries, Thomas & Kramer, 2003; Florant & Healy, 2012). However, many granivorous hibernators also utilized cached seeds for food during the hibernation period, which allows greater flexibility in use of torpor (e.g., Vander Wall, 1990; French, 2000; Humphries, Thomas & Kramer, 2003; Sheriff et al., 2011). Jumping mice are somewhat unique among granivores in that they apparently do not use food caches and hence must rely on body fat for all energy needs during hibernation (French, 1985; Humphries, Thomas & Kramer, 2003). Thus, the phenology of the annual cycle in jumping mice, with its attendant costs and benefits, may differ in comparison with other taxonomic groups.

Other than through use of radio-telemetry, there are no effective methods to detect most species while they are in hibernation, since they are inactive in underground burrows. Consequently, in order to study these animals in the wild, information must be available on when they are active above ground. This is particularly true for threatened and endangered species because populations must be monitored to detect trends, and surveys for the species’ presence are often required in areas of potential habitat when activities might cause harm to the animals if present. For instance, the New Mexico meadow jumping mouse, Zapus hudsonius luteus, was listed as endangered under the US Endangered Species Act in June 2014 due to substantial declines in populations over the last several decades (US Fish & Wildlife Service, 2014). This subspecies was originally described as a distinct species (Miller, 1911) and recent molecular analyses have validated it as relatively diverged monophyletic clade (Malaney & Cook, 2013). Zapus h. luteus occurs in the American Southwest, with an historical range that included portions of southern Colorado, New Mexico, and central and eastern Arizona (Hafner, Petersen & Yates, 1981; Frey, 2012; Malaney, Frey & Cook, 2012). It is a specialist of riparian habitats and hence its distribution includes both low elevation sites within desert biomes and high elevation sites within boreal biomes (Frey & Malaney, 2009; Malaney, Frey & Cook, 2012).

The known distribution of Z. h. luteus extends between 32.7°N and 37.2°N latitude and between 1,375 m and 2,926 m elevation, possibly as low as 935 m (i.e., Camp Verde, Yavapai County, Arizona; Frey, 2008; Frey, 2012). Given the large size and extreme topographic variability of this region, Z. h. luteus is expected to exhibit geographic variation in phenology of its hibernation and reproduction. However, there is little existing information on phenology of the annual cycle in this taxon. In addition, there are relative few studies on phenology of the annual cycle in other taxa of jumping mice. Within other subspecies of the meadow jumping mouse, Z. hudsonius, phenology has been described at four locations, three in the eastern US (Minnesota, Quimby, 1951; New York, Whitaker, 1963; Michigan, Nichols & Conley, 1982; Muchlinski, 1988) and one in Colorado (Meaney et al., 2003), but none of these studies evaluated variation in phenology at locations in different environments. Within the western jumping mouse, Z. princeps, phenology has been described for a population in Alberta (Falk & Millar, 1987), and three studies have evaluated variation in phenology of different populations living in different environments in Wyoming (Brown, 1967) and Utah (Cranford, 1978; Cranford, 1983). No studies have evaluated phenology of hibernation in the Pacific jumping mouse, Z. trinotatus (Verts & Carraway, 1998). A comparison of these studies indicates that different species and populations of jumping mice exhibit different phenologies, rendering inference to Z. h. luteus impossible. Thus, the purpose of my study was to determine the timing of hibernation and reproduction in Z. h. luteus and to compare the phenology with other taxa of jumping mice. First, although the annual cycle for most hibernators is similar, I summarize existing knowledge about the phenology of hibernation and reproduction in jumping mice, with a specific focus on Z. hudsonius, in order to establish specific assumptions to be tested.

Hibernation and reproductive phenology in jumping mice

Jumping mice (Subfamily Zapodinae) are in the Holarctic rodent Family Dipodidae, which also includes the birch mice (subfamily Sicistinae) and jerboas (subfamilies Allactaginae, Cardiocraniinae, and Dipodinae). Traditionally, Zapodinae includes five species: the Chinese jumping mouse (Eozapus setchuanus), and two genera endemic to North America, including the woodland jumping mouse (Napaeozapus insignis), and three species of jumping mice (genus Zapus; Wilson & Reeder, 2005). The genus Zapus (hence forth, jumping mice) have a boreo–montane distribution. The meadow jumping mouse (Z. hudsonius) has the largest geographic range, which extends across the boreal zone of Alaska and Canada and in the eastern US south through the southern Appalachian Mountains region (Laerm, Ford & Chapman, 1996). Z. hudsonius also occurs as isolated or semi-isolated populations in the western US, including along the east slope of the Front Range of the Rocky Mountains (Z. h. preblei) and in the American Southwest (Z. h. luteus; Malaney, Frey & Cook, 2012). The western jumping mouse (Z. princeps) occurs throughout the Rocky Mountain region, from southern Alaska south into the Sierra Nevada of California and the Southern Rocky Mountains in northern New Mexico, with an eastern extension across the northern Great Plains to Minnesota (Malaney et al., 2013). However, recent phylogenetic analyses indicate Z. princeps is represented by at least 5 clades, which may redefine species boundaries (Malaney et al., 2013). Lastly, the Pacific jumping mouse (Z. trinotatus) is restricted to the Sierra Nevada and Pacific coastal area from southern Canada south into California (Wilson & Reeder, 2005). Malaney et al. (2013) found that Z. princeps is paraphyletic with respect to Z. trinotatus, but no taxonomic conclusions were made.

In jumping mice, emergence from hibernation in the spring is cued by soil temperature (Cranford, 1978; Muchlinski, 1988; French & Forand, 2000). Studies of Z. hudsonius in the eastern US have demonstrated that males emerge at lower soil temperatures than females and hence males are active above ground prior to females (Muchlinski, 1988; French & Forand, 2000). In Z. hudsonius from Ingham County, Michigan, first emergence of females averaged 14 days after first emergence of males, and the mean date of emergence of females was 17 days later than males (data from Muchlinski, 1988). Timing of emergence in Z. hudsonius is known to vary annually and geographically due to variation in soil temperature (Quimby, 1951; Muchlinski, 1988). Similarly, timing of emergence from hibernation in the western jumping mouse (Z. princeps), also is cued by soil temperature (Cranford, 1978) and hence it varies with elevation (Brown, 1967; Cranford, 1983). In Wyoming, emergence of Z. princeps occurred approximately 2 weeks later for each 305 m increase in elevation (Brown, 1967), but Cranford (1983) also observed considerable variation due to habitat quality and other local features such as aspect and shade. Female Z. princeps in Wyoming emerged 9 to 12 days later than males (Brown, 1967), but Cranford (1983) found that timing of emergence in Utah was uniform except at the highest elevations.

Evidence suggests that photoperiod cues immergence into hibernation by Z. hudsonius in the eastern US (Neumann & Cade, 1964; Muchlinski, 1978; Muchlinski, 1980). Because timing of immergence is cued by photoperiod, entrance into hibernation by adult Z. hudsonius may be more uniform both geographically and annually, in comparison to emergence. Hibernation is preceded by a critical ca 2 week period of rapid weight gain (e.g., 8% per day; Quimby, 1951; Morrison & Ryser, 1962; Muchlinski, 1988). In the eastern US adults undergo weight gain during late August and all have entered hibernation by about the end of the first week in September (Muchlinski, 1988). Adult males enter hibernation first, followed by adult females; juveniles enter hibernation last with timing dependent on birth date (later litters entering hibernation as late as October; Muchlinski, 1988). In contrast, immergence in Z. princeps was thought to be cued by availability of seeds in the diet rather than photoperiod (Cranford, 1978). This difference may be a strategy that allows Z. princeps to cope with a much shorter period of above ground activity in the high elevation sites it occupies (i.e., ca 2.7–4 months as compared with ca 5.5 months in many eastern US populations of Z. hudsonius; Cranford, 1978; Cranford, 1983; Muchlinski, 1980). During years with late spring emergence and plant growth, there might not be enough time for jumping mice to accumulate fat reserves if immergence was consistently cued by day length. Consequently, cueing on availability of seeds is thought to allow Z. princeps to initiate hibernation when conditions are most favorable (Muchlinski, 1980).

One of the main constraints of a short activity season is the number of litters that can be produced annually. Mating apparently occurs soon after the females have emerged from hibernation (Whitaker, 1972) and hence timing of spring emergence influences the time available for females to raise young to weaning and then for young of the year to mature and ultimately gain fat in preparation for hibernation. Variation in emergence times in Z. hudsonius has resulted in litters being produced 2–3 weeks later in some years (Muchlinski, 1988). Gestation is 18–21 days (Quimby, 1951) and it then requires ca 4 weeks after birth before the young are weaned and become independent (Whitaker, 1972). It required ca 90 days for a juvenile jumping mouse to attain a mass of 20 g, which is adult size (Quimby, 1951). Preparation for hibernation then requires a 2-week period of fattening (Morrison & Ryser, 1962). Thus, the minimum time required from conception to hibernation in Z. hudsonius is ca 125 days. In Z. hudsonius from the eastern US the active interval, which is the number of days from emergence of the first animal in the spring to the immergence of the last animal in fall, is 162–165 days (Muchlinski, 1988). Evidence suggests that female Z. hudsonius must achieve a large body mass in order to reproduce, which can result in delays in reproduction (Falk & Millar, 1987). Thus, some females produce their first litter within a month after emergence (i.e., early breeding females), while females that did not breed during the first month may produce a litter during the second month after emergence (i.e., late breeding females; Quimby, 1951). In the eastern US , both early and late breeding females may produce a second litter, young of early litters may breed, and there is some evidence that lactating females can become pregnant (Quimby, 1951; Nichols & Conley, 1982). Hence, in the eastern US there are two (sometimes three) peaks in reproduction during the active season and individual females may produce two and conceivably three litters per year (Quimby, 1951; Whitaker, 1963; Nichols & Conley, 1982). Similarly, both early and late litters were detectable in Z. h. preblei, even though it has a shorter active season (150 days; Meaney et al., 2003). In Z. princeps, females that emerge from hibernation with low body weights (i.e., young of prior breeding season) are more likely to delay reproduction and have smaller litters (Brown, 1967; Falk & Millar, 1987). Compared to other demographic groups, young of late litters must extend their activity season later into the fall in order to gain weight; even so, these young ultimately have lower survival likely due to a longer period of exposure to predators and lower body weights when entering hibernation (Muchlinski, 1988; Meaney et al., 2003; Schorr, Lukacs & Florant, 2009).

On basis of observed patterns in phenology of hibernation and reproduction in other jumping mice, I expected for Z. h. luteus that: (1) emergence from hibernation in spring will be later for higher latitudes and higher elevations due to overall cooler climate and hence cooler soil temperatures; (2) immergence into hibernation will be later for lower latitudes and lower elevations due to longer growing seasons; (3) the active interval will be shorter for montane populations as opposed to valley populations, and (4) the number of litters possible will be reduced for montane populations. I also report an apparent midsummer hiatus in above ground activity by Z. h. luteus at Bosque del Apache National Wildlife Refuge (BANWR), which is the lowest elevation and warmest site known to be currently occupied by the taxon.

Methods

Currently, field studies to examine phenology of the few remaining populations of Z. h. luteus are not generally feasible because such studies require trapping animals and the populations are small and at high risk of extinction (US Fish & Wildlife Service, 2014). Consequently, I extracted data from museum specimens to supplement the limited information available in published literature (Zwank, Najera & Cardenas, 1997; Wright & Frey, 2015), unpublished theses (Najera, 1994; Wright, 2012), and agency reports (JL Morrison, 1987, unpublished data; JL Morrison, 1988, unpublished data; SR Najera, PJ Zwank, & M Cardenas, 1994, unpublished data; Wright & Frey, 2011; Frey & Wright, 2012; BANWR, 2014). This included all known specimens of Z. h. luteus with recorded dates of capture (N = 309) and represented the taxon’s current geographic range (Table S1). Specimens were captured by me (N = 63) or were in the following museum collections: Academy of Natural Sciences of Philadelphia (ANSP; N = 10); Arizona State University Mammal Collection (ASUMC; N = 6); Denver Museum of Natural History (DMNH; N = 14); University of Kansas, Museum of Natural History (KU; N = 5); Museum of Northern Arizona (MNA; N = 7); University of New Mexico, Museum of Southwestern Biology (MSB; N = 118); University California, Berkeley, Museum of Vertebrate Zoology (MVZ; N = 11); New Mexico Museum of Natural History and Science (NMMNHS; N = 2); New Mexico State University, Vertebrate Collection (NMSU; N = 6); San Diego Natural History Museum (SDNHM; N = 25); Museum of Texas Tech University (TTU; N = 1); University of Arizona, Collection of Mammals (UA; N = 9); University of Illinois Museum of Natural History (UIMNH [collection transferred to MSB]; N = 6); University of Utah, Utah Museum of Natural History (UMNH; N = 7); United States National Museum (USNM; N = 18); Western New Mexico University (WNMU; N = 1).

Table 1 Table of records of jumping mouse specimens by month.

Percent of records by month for the New Mexico meadow jumping mouse (Zapus hudsonius luteus) based on museum specimens from low elevation valleys and montane populations. Percent of records by month for Bosque del Apache National Wildlife Refuge (Valley, Rio Grande population) is based on field data reported by Najera (1994) and SR Najera, PJ Zwank, & M Cardenas (1994, unpublished data).

	Bosque del Apache (N = 78)	Valley (N = 67)	Montane (N = 242)	
May	41.0	4.5	0.0	
June	28.2	14.9	11.6	
July	17.9	29.9	44.2	
August	0.0	25.4	30.2	
September	3.8	25.4	12.8	
October	9.0	0.0	1.2	

Because soil temperature is influenced by elevation and latitude, I categorized specimens into nine populations divided into two groups: montane (N = 242; Sangre de Cristo Mountains, Jemez Mountains, Sacramento Mountains, White Mountains) and valley (N = 67; Florida River, Sambrito Creek, Mora River, Rio Chama, Rio Grande). I constructed histograms of numbers of specimens of each sex by Julian date to evaluate times of emergence from and immergence into hibernation. The Julian date represents the day a specimen was captured. Because trapping does not necessarily detect either the first or last above ground activity of a population, dates of emergence and immergence should be considered conservative estimates. I examined timing of emergence according to relative temperature equivalents of locations. To a large extent, the climate of a location is ultimately based on its latitude and elevation (i.e., higher latitudes and higher elevations have cooler temperatures). Consequently in order to compare locations that vary in latitude and elevation I calculated a temperature equivalent for locations based on the method described in Frey, Yates & Bogan (2007). The temperature equivalent was set in relation to a hypothetical location located 34°N latitude, 1,981 m elevation with a mean annual temperature of 12.2 °C, which are approximate averages for New Mexico. The temperature lapse rate was set to 0.56 °C per 1° latitude and per 76.2 m elevation. The temperature equivalent (in Celsius) for a location was calculated: TE = 12.2 + ([1,981 − elevation in meters]*[0.556/76.2]) + ([34 − degrees latitude]*0.556).

For museum specimens, I examined cranial and dental characters when possible to establish relative age. Specimens were assigned to 1 of 6 age classes according to wear on the cheekteeth as described by Krutzsch (1954) and specimens were assigned to 1 of 8 age classes based on eruption and wear on the third upper molar (M3) and closure of the basioccipital-basisphenoid suture according to Jones (1981). The Krutzsch (1954) and Jones (1981) age classes for a specimen were transformed into fractions of the total age class possible (e.g., a Krutzsch age class 4 = 4/6 = 0.66). Following Frey (2008), the age class was the mean of the two fractions for an individual. No studies have correlated age classes based on cranial and dental features with known age individuals. Consequently, based on an examination of age classes of animals by date of capture (Fig. 1), I distinguished between two age groups: young of the year and adult. All animals immerging from hibernation were at age class ≥0.35. During hibernation growth essentially ceases and no wear on teeth occurs because jumping mice are not eating. Consequently, I considered age class 0.35 to include both older young of the year animals entering hibernation at the end of their first growing season, and adults emerging from hibernation during their second active season (i.e., first breeding season). I considered animals at age class >0.35 to be older, but of unknown age. Animals in age class <0.35 were considered young of the year, with animals in age class ca 0.2 considered recently weaned juveniles.

Figure 1 Figure of age class of jumping mouse specimens by date of capture.

Age class of specimens of the New Mexico meadow jumping mouse (Zapus hudsonius luteus) by date of capture for (A) valley populations and (B) montane populations. Age class was determined by characteristics of the skull and dentition.

For museum specimens, the primary data I used to establish timing of reproduction was evidence of pregnancy via counts of embryos recorded on specimen tags because these data are unequivocal. I did not consider specimen tag data on lactation reliable because “lac” may be recorded for females in a variety of conditions such as swollen mammae or post-lactation. I used specimen data on reproductive condition of males only to supplement the pregnancy data because data on scrotal versus nonscrotal testes were sparse. I used a 21-day gestation period (though gestation actually may vary from 18–21 days), a 28-day nesting period (time from parturition to weaning and independence), and growth rates for young of the year animals to back-calculate dates of conception and parturition, though it is cautioned that back-calculated dates based on mass of juveniles are often overestimated (Quimby, 1951).

Problems identifying reproductive state during field studies

I found inconsistencies in the recording of pregnancy in field data. Najera (1994) and Wright (2012) reported jumping mice of all ages, including juveniles and pregnant females, during May at BANWR. However, evidence suggests that some assignments of age and reproductive status were incorrect. First, while some field studies of jumping mice have used body mass as an indicator of age (e.g., Brown, 1967; Nichols & Conley, 1982), body mass is highly variable within individual jumping mice and overwintered adults can emerge from hibernation with relatively low body mass (<20 g; Quimby, 1951; Meaney et al., 2003). As an example, Wright (2012) captured a jumping mouse that was unequivocally a juvenile (7 g and juvenile pelage) on 16 August 2009. This animal was recaptured the following spring on 18 May 2010, at which time it weighed only 14 g, which equates to a 60 day old juvenile (Quimby, 1951)! Hence, body weight alone cannot be used to determine age, at least in the early part of the active season (Meaney et al., 2003).

Second, evidence suggests that some field evaluations of pregnancy may be inaccurate. For instance, Wright (2012) reported capturing 9 females that were evaluated as pregnant (Table S2). Of those, 7 had excessive body weight (>22 g; Meaney et al., 2003) or enlarged mammae that corroborated pregnancy. In addition, 7 were radio-tracked. Three exhibited normal activity behaviors during the radio-tracking session, which consisted of nightly foraging in herbaceous wetland habitats and nesting during the day in above-ground nests in grasses (Wright & Frey, 2015). In contrast, four exhibited dramatically different behaviors that were interpreted as tending a maternal nest with nursing young, although the possibility that these females had entered hibernation cannot be discounted. These females left their typical wetland habitats and became almost entirely inactive for ca 2 or more weeks in underground burrows located in woody habitats devoid of herbaceous vegetation. Ryon (2001) described a similar burrow that was used as a maternal nest by Z. h. preblei. Importantly, no females captured in May and recorded as pregnant had corroborating evidence of pregnancy. Therefore, it is possible that fat layers remaining from hibernation made the females appear pregnant when they were not. Consequently, field evaluations of pregnancy should be suspect without corroborating information such as swollen mammae, palpated fetuses, excessive weight, or behavioral changes. Thus, for field data I only considered females as pregnant if they were recorded as pregnant and there were other data corroborating pregnancy such as excessive body mass (>22 g), which is consistent with late pregnancy (Quimby, 1951; Meaney et al., 2003).

Results

Emergence from hibernation

Emergence from hibernation was earlier for some valley populations than montane populations (Fig. 2). The earliest captures represented by the specimens were 3 males on 24 May from the Rio Grande valley population (Isleta Pueblo; 34.9°N latitude, 1,495 m elevation). However, field studies recorded slightly earlier dates. Further south along the Rio Grande at Bosque del Apache National Wildlife Refuge, Socorro County (BANWR; 33.8°N latitude, 1,370 m elevation), Najera (1994; see also Zwank, Najera & Cardenas, 1997 trapped for Z. h. luteus beginning in March 1992, but did not capture a jumping mouse until 13 May. Males made up 83% of captures during May with the first female caught on 20 May (Najera, 1994). The report of a capture on 13 March by Zwank, Najera & Cardenas (1997) is an error (see Table 15 and Appendix B in Najera, 1994). During another study at BANWR during 2009–2011, trapping started on 13 May with the first captures (two males) on 18 May 2010; the first female that year was not caught until 18 June (Wright, 2012; Frey & Wright, 2012). During the previous year, trapping started 21 May with a male caught on 22 May and the first female caught on 26 May; over both years 78% (N = 9) of jumping mice caught in May were male. The earliest capture date for a valley specimen outside the middle Rio Grande valley were two females caught 24 June at Espanola, Rio Arriba County, New Mexico (36.0°N latitude, 1,700 m elevation).

Figure 2 Histogram of male and female jumping mouse capture dates by week.

Activity season of male (black bars) and female (white bars) New Mexico meadow jumping mice (Zapus hudsonius luteus) from (A) valley populations and (B) montane populations, based on dates of capture recorded on museum specimen labels. Julian date equivalents are 121, 1 May; 152, 1 June; 182, 1 July; 213, 1 August; 244, 1 September; 274, 1 October; 305, 1 November.

No specimens from montane areas have been captured in May (Table 1). The earliest capture represented by specimens from montane areas was a male caught 11 June at Sugarite Canyon, Las Animas County, Colorado (37.0°N latitude, 2,300 m elevation; Jones, 1999). Earliest dates of specimens in other well-sampled montane populations include 18 June (Sacramento Mountains: Tularosa Creek, Otero County, 33.1°N latitude, 2,050 m elevation), 20 June (White Mountains: West Fork Black River, 33.8°N latitude, 2,330 m elevation), and 28 June (Jemez Mountains: San Antonio Creek, Sandoval County, 35.9°N latitude, 2,355 m elevation). Of 12 montane specimens with June capture dates and gender data, only 4 were females, which were taken 18, 27, 29, and 30 June. Morrison (1987, unpublished data) conducted the only field study that attempted to determine timing of emergence in a montane population at Fenton Lake in the Jemez Mountains, Sandoval County, New Mexico (35.9°N latitude, 2,350 m elevation). Her first capture was a male on 13 June with the first female not captured until 27 June (JL Morrison, 1987, unpublished data). She caught a total of 14 males and 1 female during June (JL Morrison, 1987, unpublished data).

The relationship between temperature equivalents of locations and known first emergence dates predicted that for each degree Celsius increase in temperature equivalent the emergence date would occur more than three days earlier (Fig. 3). Dates of earliest known museum specimens from specific locations often were later than predicted, likely due to small sample sizes and because specimens were collected incidentally without special attempt to determine emergence date. Predicted dates of first emergence for key populations of Z. h. luteus extend over 47 days from 6 May to 22 June (Table 2). It should be noted that variation will exist around these predicted dates due to small sample sizes, coarse nature of the model, and annual and site-specific variation in soil temperature.

Figure 3 Figure of relationship between temperature equivalents and date of first emergence.

Relationship between the temperature equivalent (C°) of a location and the earliest known date for emergence of the New Mexico meadow jumping mouse (Zapus hudsonius luteus) from hibernation. Temperature equivalents are relative to a hypothetical location with approximate average conditions for New Mexico: 34° N latitude, 1,981 m elevation, and mean annual temperature of 12.2 °C. Solid dots and regression line are based on dates of earliest capture during field studies; stars are earliest dates of representative museum specimens. Julian dates range from 1 May (121) to 3 July (184).

Table 2 Table of predicted dates of first emergence.

Latitude, longitude, elevation, temperature equivalent, and predicted date of first emergence of the meadow jumping mouse (Zapus hudsonius luteus) from hibernation. The temperature equivalent is a relative measure of mean annual temperature that corrects locations for latitude and elevation according to approximate means in New Mexico and a temperature lag rate of 0.56 °C per 1° latitude and 76 m elevation. The predicted date of first emergence is based on the regression equation in Fig. 3.

Population	North latitude (degrees)	West longitude (degrees)	Elevation (m)	Temperature Equivalent (C°)	Predicted date of first emergence	Earliest known dates	
Verde River, Camp Verdea	34.6	111.8	950	19.4	6 May		
Rio Grande, Bosque del Apache	33.8	106.9	1,370	16.8	14 May	13 May, 18 May	
Rio Grande, Isleta	34.9	106.7	1,495	15.2	18 May	24 May	
Rio Grande, Espanola	36.0	106.1	1,700	13.1	25 May	24 Jun	
White Mountains, Campbell Blue Creek	33.7	109.1	2,000	12.2	28 May		
Sacramento Mountains, Tularosa Creek	33.1	105.7	2,050	12.2	28 May	18 June	
Sacramento Mountains, Rio Penasco	32.8	105.6	2,170	11.5	30 May		
Piedra River, Sambrito Creek	37.0	107.5	1,860	11.5	30 May	21 Mayb	
Florida River, Florida	37.2	107.7	2,050	9.9	4 June		
White Mountains, West Fork Black River	33.8	109.4	2,330	9.8	5 June	20 June	
Mora River, Mora	36.0	105.3	2,185	9.6	5 June		
White Mountains, Nutrioso Creek	33.9	109.2	2,450	8.8	7 June		
Sangre de Cristo Mountains, Fort Burgwin	36.3	105.6	2,250	9.0	7 June		
Jemez Mountains, Fenton Lake	35.9	106.7	2,350	8.5	9 June	13 June	
White Mountains, North Fork White River	34.0	109.7	2,500	8.4	9 June	24 June	
Jemez Mountains, San Antonio Creek	35.9	106.6	2,355	8.4	9 June	28 June	
Sacramento Mountains, Aqua Chiquita Creek	32.7	105.7	2,600	8.4	9 June		
Sangre de Cristo Mountains, Sugarite Canyon	37.0	104.4	2,300	8.2	9 June	11 June	
Sangre de Cristo Mountains, Coyote Creek	36.2	105.2	2,365	8.2	10 June		
Sacramento Mountains, Wills Canyon	36.8	105.7	2,680	7.8	11 June		
Sangre de Cristo Mountains, Rito la Presac	36.1	105.5	2,670	6.0	16 June		
White Mountains, Lee Valley Creek	33.9	109.5	2,880	5.7	17 June		
Sangre de Cristo Mountains, Rio Hondod	36.6	105.4	2,870	4.3	22 June		
Notes.

a See Frey (2012) for information about a population in the Verde River watershed.

b Trapping started 20 May; JL Zahratka, pers. comm., 2015.

c See Frey (2008) for information about this location.

d See Hafner, Petersen & Yates (1981) and Frey (2008) and for information about this population.

To summarize available information on emergence dates for Z. h. luteus, valley populations emerge from hibernation in mid-May, with the earliest above-ground activity recorded on 13 May at BANWR. This early date of emergence is fairly reliable because trapping had started earlier in an attempt to determine emergence. The only similar field study in a montane population (Fenton Lake) documented the earliest above-ground activity on 13 June. In other montane populations, dates of capture for museum specimens were as early as 11 June at Sugarite Canyon. Although details about trapping dates are not available for the Sugarite Canyon example, I consider this date fairly accurate because trapping began in May and the 11 June date was represented by the capture of two nonscrotal males (Jones, 2002). In both valley and montane populations, males emerge from hibernation first; males make up the majority of captures during May and June, for valley and montane populations respectively.

Immergence into hibernation

Among specimens from low elevation valley populations, none had capture dates after 16 September (Table 1 and Fig. 2). Data on age class revealed that older adult age classes disappeared by 4 September, while younger animals disappeared by 16 September (Fig. 1). However, field studies reveal later dates of immergence. During a 7–19 September trapping period in the Rio Chama and Rio Grande valleys near Espanola, Rio Arriba County, Morrison (1988, unpublished data) caught two males with weights (31.5 g and 37.0 g) that are typical of adult jumping mice imminently ready to enter hibernation.

At BANWR, Najera (1994 see also Zwank, Najera & Cardenas, 1997) caught jumping mice through September and until 22 October, although all were considered young of the year in these months except an adult female (24.0 g) on 12 September and an adult male that weighed 32.0 g on 27 September and 35.0 g on 1 October, and was imminently ready for immergence (Table 3). Wright (2012) did not catch any jumping mice at BANWR in September, but caught a 20.0 g young of the year female on 22 and 25 October. This jumping mouse was radio-collared and its last above ground movement was 26 October (Wright, 2012). No jumping mice have been detected above ground at BANWR between 27 October and 13 May, despite 1,740 trap-nights (1 trap-night = 1 trap set for 1 night) by Najera (1994) in March, April and November and an effort of 7,540 trap-nights during relatively warm spells throughout this period (Wright & Frey, 2011; Wright, 2012).

Table 3 Table of age of jumping mice by month at BANWR.

Age of New Mexico meadow jumping mice (Zapus hudsonius luteus) by month captured in the middle Rio Grande valley at Bosque del Apache National Wildlife Refuge, Socorro County, New Mexico. The number of known pregnant females is indicated with a “p” in parentheses.

	Museum specimensa	Najera (1994); SR Najera, PJ Zwank, & M Cardenas (1994, unpublished data)b,d	Wright (2012); Frey & Wright (2012)c,d	
Month	Age class < 0.35	Age class = 0.35	Age class > 0.35	Young of year	Adult	Young of year	Adult	
May	0	0	0	0	34	0	10	
June	0	0	4	0	26 (p = 1)	0	7 (p = 1)	
July	0	2	3	1e	14 (p = 5)	0	11 (p = 6)	
August	1	9	1	0	0	2	0	
September	2	8	2	1	2	0	0	
October	0	0	0	7	0	1	0	
Notes.

a All museum specimens were collected 1976–1979. Fifteen of the specimens were found drowned in wading pools that had been set up for a toad behavioral study in 1977–1978 (DJ Hafner, pers. comm., 2007). Hence, recorded dates might be later than actual date of death. Aging of specimens was via cranial and dental characters as described in the text. Age class <0.35 were considered young of the year; age class 0.35 were considered older young of the year immerging into hibernation and emerging from hibernation during their second active season; age class >0.35 were older adults.

b Results are combined for data collected June–October 1991 and May–July 1992.

c Results are combined for May–August 2009 and May–October 2010.

d Following Nichols & Conley (1982) and Meaney et al. (2003) all May and June individuals were assumed to be adults (i.e., overwintered), regardless of body mass. For July, it was assumed that independent young could first appear on 11 July (based on average date of female emergence plus 49 days for gestation and nursing) at which time they weigh ca 8–10 g. I regarded any animal <14 g (i.e., lowest known weight of an adult in spring) in July as young of the year. For August–October, I used the relation illustrated by Meaney et al. (2003) between date and body mass of Zapus hudsonius preblei to distinguish age classes.

e An 8 g juvenile male caught on 31 July.

Among specimens from montane populations, only 14% were captured after August (Table 1). Of specimens of known age, older adults were not detected above ground after 7 September, and the last recorded capture was 19 September (Fig. 1). However, three specimens of unknown age class were caught in October, including 4, 19, and 26 October, all from the White Mountains, Arizona (Table 1). At Fenton Lake, Morrison (1987, unpublished data) caught 5 jumping mice in October that previously had been captured with the last caught on 3 October after which trapping ended. Thus, there could have been later dates of activity. Morrison (1987, unpublished data) thought that most adults had entered hibernation by mid-September and that later occurrences were young of the year. However, in the White Mountains near Greer, I caught five jumping mice on 12 September, which included four young of the year weighing <20 g, and one adult female that weighed 34 g that was imminently ready for hibernation.

To summarize available information on immergence dates for Z. h. luteus, in both valley and montane populations older adults disappear from the trappable population during early September, while young of the year do not disappear until late October. Among adults, most above ground activity during the later part of the active season appears to be females, perhaps representing individuals that reproduced. Due to limitations of the available data, it is not possible to determine when animals begin immergence. It is possible immergence of adults could occur as early as August, as occurs in some adult Z. h. preblei (Meaney et al., 2003). Research on some other species of mammals has shown that local climate conditions may influence whether individuals hibernate or not during winter (e.g., Davis, 1976; Lehmer et al., 2006) and it has been speculated that Z. h. luteus may not hibernate at BANWR given the relatively warm climate (JL Morrison, 1988, unpublished data; JL Morrison, 1988, unpublished data; SR Najera, PJ Zwank, M Cardenas, 1994, unpublished data; Najera, 1994). However, based on a lack of detectable above-ground activity during the winter, jumping mice at BANWR probably do hibernate and the population had an active interval of 162 days (= ca 5.5 months) in 2010 (Wright, 2012). The active interval for montane populations cannot be as precisely determined, but is ca 124–135 days (ca 4 to 4.5 months) for Fenton Lake (13 June to probably mid-October).

Reproduction at BANWR

The most well studied population of Z. h. luteus is at BANWR. At this location, most pregnancies occur in late June and July. Wright (2012) caught females with corroborating evidence of pregnancy on 23 June, and 20, 25, 26, and 27 July (Table S1). For radio-collared jumping mice, parturition dates probably coincided with retreat to underground burrows on 28 and 30 July (Table S1). Hence, back calculated conception dates are 7 and 9 July, and juveniles from those litters would be weaned and become part of the trappable population on 25 and 27 August, respectively. There may be some variation in these dates because some of the radio-collared pregnant jumping mice had unusual and reduced activity just prior to retiring in the underground burrows and it is conceivable that they gave birth outside the burrows and then moved the young or conversely they retired to the burrow just prior to giving birth. Data from Najera (1994) support late June through July as the primary period for pregnancy at BANWR. A 29.0 g female appeared pregnant on 9 July and was recaptured at least 7 days later (no date given) and appeared to be no longer pregnant but lactating (Najera, 1994). Females captured that were evaluated as both pregnant and with large body mass (>22 g) included two on 15 July 1991 (25.0 and 29.0 g), and one each on 27 June 1992 (28.0 g), 8 July 1992 (22.5 g), 9 July 1992 (29.0 g), and 16 July 1992 (26.0 g) (Najera, 1994).

The earliest date of pregnancy recorded at BANWR is a 21.5 g female was captured on 15 June 2014 that was confirmed pregnant through palpation of embryos and presence of enlarged nipples (BANWR, 2014). Two additional female jumping mice (23.5 g and 26.5 g) were caught on 19 June and confirmed pregnant by palpation of fetuses, and presence of enlarged nipples and vulva (BANWR, 2014). In addition, Najera (1994) caught an 8 g juvenile male (age 21–30 days according to Quimby, 1951) on 31 July, that likely would have had been conceived 10–19 June (Najera, 1994). Given that there is a lag between conception and ability to detect pregnancy, pregnancies may occur as early as the first week of June at BANWR.

Najera (1994) and Zwank, Najera & Cardenas (1997) suspected that breeding at BANWR took place as late as August because they caught young of the year animals in October. However, this estimate of breeding date may be incorrect. The size range of jumping mice they caught in October was 15.0 to 24.5 g (mean 18.5 g). These included a 15.5 g female on the last date (22 October) jumping mice were caught. According to Quimby (1951), this female was approximately 70 days old and hence it had a back-calculated parturition date of 13 August and conception date of 23 July. Thus, no breeding (i.e., conception) is verified after July at BANWR, though some females may not give birth until early August. Similarly, at BANWR males with scrotal testes were captured most frequently in June and July, with a smaller proportion in May; none were found after July (Najera, 1994; Wright, 2012).

No museum specimens taken at BANWR had data about embryos. A large series of specimens collected at BANWR had been salvaged, apparently drowned, from wading pools that were being used for amphibian experiments in 1977 (DJ Hafner, pers. comm., 2007). Those specimens were not used for these analyses because of uncertainty about when each specimen died. Of the remaining specimens, an adult female captured on 22 July and two adult females captured on 2 September were recorded as possessing uterine scars and lactating, suggesting that they had recently or were currently nursing young in a nest.

To summarize reproductive information for BANWR, some males start to become reproductively active in May, with higher proportions becoming reproductively active in June and July (Fig. 4). Pregnant females are known from 15 June to 27 July. Other evidence suggests conception during the first and second week of June. There is no convincing evidence for pregnancies in May and no reproductive activity is verified for later than 25 August (Table S1). Independent young first appear in August. However, it should be cautioned that these dates are conservative given the small sample sizes. The earliest verified date of capture for a young of the year is 31 July.

Figure 4 Schematic of timing of hibernation and reproduction.

Generalized schematic of the timing of key life history events for the New Mexico meadow jumping mice (Zapus hudsonius luteus) at (A) Bosque del Apache National Wildlife Refuge (BANWR), and (B) in montane populations. Solid lines represent time frames documented by observation; dashed lines represent time frames that are inferred based on timing of other observed events. Asterisks indicate pregnant females captured at other valley locations (Sambrito Creek and Isleta) that suggest a wider possible time frame for pregnancies at BANWR.

Reproduction in other populations

Information from other valley populations indicates a broader time range for pregnancy as compared with data from BANWR (Fig. 4). For instance, data indicate pregnancies can occur in early June. A specimen caught on 13 June from the Rio Grande valley near Isleta was carrying five embryos in an early development stage (<5 mm long). Similarly, Morrison (1988, unpublished data) caught a 13 g juvenile on 1 August in the Rio Grande valley near Casa Colorada Wildlife Area, Valencia County; assuming it had recently been weaned, the back-calculated conception date was prior to 13 June. Along a western tributary of Sambrito Creek, Archuleta County, Colorado, a 16.5 g juvenile (age class 0.18) male was caught on 25 July; assuming it had recently been weaned, the back-calculated conception date was prior to 6 June. Data also indicate later pregnancy into August. Near Isleta, Valencia County, Morrison (1988, unpublished data) caught a female (age class 0.44) on 19 August that had enlarged mammae and was carrying seven embryos. Based on age class data, young of the year enter the trappable population about 25 July (Fig. 1).

Dates for pregnancies in montane populations are generally later than in valley populations (Fig. 4). At Fenton Lake in the Jemez Mountains, Morrison (1987, unpublished data) evaluated males as scrotal between 23 June and 18 July, females as pregnant between 28 July and 15 August, and females having enlarged mammae between 21 July and 29 August. However, in the Jemez Mountains I captured a 22 g female on 1 July that was carrying six 2 mm embryos and a 18.5 g female on 5 July that was carrying six 6 mm embryos. Females specimens with embryos were captured 22 July–16 August (average date 27 July; N = 9) in the White Mountains and 15 July–17 August (average date 26 July; N = 9) in the Sacramento Mountains. Based on age class data, young of the year in montane areas enter the trappable population about 17 August (Fig. 1).

To summarize, reproductive data for populations other than BANWR are sparse and mostly provide a range of dates based on confirmed or inferred reproductive status. Data are particularly sparse for other valley populations, where pregnancies are inferred to have occurred as early as the first week in June based on back-calculated dates. It is assumed that most pregnancies occur during June and July as at BANWR. However, there is a single record of a pregnancy from as late as 19 August. This female was an older adult and hence this represents an exceptionally late litter. For montane populations, pregnancies occur at least 2 weeks later than generally occur at BANWR, with confirmed pregnancies documented from 1 July to 17 August.

Bosque del Apache summer activity hiatus

BANWR is the southernmost location for Z. h. luteus along the Rio Grande and the lowest elevation location (i.e., highest temperature equivalent) where the species is currently known to persist. Field studies at BANWR have revealed a sharp reduction in detectable above-ground activity of jumping mice during late summer. In 1991 and 1992, Najera (1994) caught jumping mice in June, July, September and October, but caught none 16 July–10 September, which included a sampling effort of 4,708 trap-nights in August (Najera, 1994). In 2009 and 2010, Wright (2012) captured only a juvenile male on 16 August during a 30 July–17 August 2009 trapping period with an effort of 2,910 trap-nights and a 16 g male on 28 August during a 23 August–20 September 2010 trapping period with an effort of 4,320 trap-nights (Table 3).

Discussion

Zapus h. luteus exhibits geographic variation in phenology of key life history events. However, not all expectations were observed and the phenology of hibernation and reproduction in Z. h. luteus was fundamentally different compared to other subspecies of Z. hudsonius (Fig. 4). As expected, montane populations experienced later first emergence from hibernation in comparison with valley populations. However, contrary to expectations, last immergence into hibernation occurred at about the same time for both valley and montane populations, though immergence dates were difficult to precisely define due to data limitations. Consequently, as expected, the active interval was shorter for montane populations in comparison with valley populations and the number of litters possible per year was reduced from conceivable two per year in valley populations to only one per year in montane populations. Among populations of Z. h. luteus, the population occurring at BANWR, which was the warmest location, had a phenology most similar to other subspecies of Z. hudsonius. The active interval at BANWR (162 days) was the same as for Z. hudsonius from central Michigan (Muchlinski, 1988). However, the timing differed. Jumping mice at BANWR emerged ca 4 weeks later than those in Michigan, which usually emerge in late April (Nichols & Conley, 1982; Muchlinski, 1988). Rather, the timing of emergence in Z. h. luteus at BANWR (13 May, 18 May) was similar to Z. h. preblei from foothills of the Rocky Mountain Front Range (19 May), although Z. h. luteus remains active longer in the fall compared to Z. h. preblei (last recorded above ground activities were 15 October and 26 October for Z. h. preblei and Z. h. luteus, respectively) resulting in a slightly longer active interval for Z. h. luteus (Meaney et al., 2003). Prior studies have indicated that emergence in jumping mice is cued by soil temperature (Cranford, 1978; Muchlinski, 1988; French & Forand, 2000), which is a product of the radiation regime, moisture content, and snow cover of a location (Baker, 1971). Although soil temperatures are not available, the mean date of last spring freeze (ca 10 May; The Climate Source, 2000) is similar for BANWR, the Front Range, and central Michigan. Thus, the stark difference in emergence dates between subspecies of Z. hudsonius from the Rocky Mountain region and the eastern US might relate to an unknown environmental variable or to adaptive evolutionary differences.

The greatest deviation from the typical active season pattern of Z. hudsonius in the eastern US was observed for montane populations of Z. h. luteus (Fig. 4). Montane populations of Z. h. luteus had an active interval of ca 124–135 days, which was reduced by ca 4–5.5 weeks compared with eastern US Z. hudsonius (162 days) and by ca 2–3.5 weeks compared with Z. h. preblei (150 days). Rather, the active interval was similar to populations of Z. princeps occurring at similar elevations (2,591 m) but further north in Wyoming (124 days; Brown, 1967). However, again, the timing was different. In part, Z. princeps copes with the short summers of high elevation montane sites by hibernating early, with the last individuals captured in early to mid September (earlier at higher elevations). In addition, Z. princeps emerges from hibernation 4–8 weeks later than eastern US Z. hudsonius. At the lowest elevations (2,591) Z. princeps emerges from hibernation in mid May (16 May), nearly identical to the population of Z. h. luteus at BANWR (Brown, 1967). However, populations of Z. princeps at 2,896 m emerge on 1 June, and those at 3,200 m on 13 June (Brown, 1967). Thus, montane populations of Z. h. luteus emerge from hibernation as late as very high elevation populations of Z. princeps. But, unlike the truncated activity period in fall experienced by Z. princeps, young of the year Z. h. luteus from montane locations may not enter hibernation until October, like Z. h. preblei (Meaney et al., 2003). Thus, different species and populations of jumping mice adjust to short activity intervals in different ways.

The length of the active interval dictates the maximum duration of reproduction possible for a population (active interval should not be confused with the above-ground activity period of individuals; adults might have activity periods of <90 days). Although data are sparse, there is no evidence for more than one litter per year in any population of Z. h. luteus. For montane populations of Z. h. luteus there is a single peak in pregnancies during late July. As in Z. princeps, which has a similarly short active interval, it seems unlikely that more than one litter per year is possible for Z. h. luteus in montane locations (e.g., Brown, 1967; Falk & Millar, 1987). To do so, a female would have to conceive the first litter immediately upon emerging from hibernation. She then would have to conceive the second litter while still nursing the first litter. Finally, the young of the second litter would likely have fewer than three weeks following weaning to reach adult size and prepare for hibernation. Such a scenario seems improbable. Data on other jumping mice have shown that younger and lower weight females delay pregnancy, with possibly only females that have achieved their third active season and that emerged from hibernation with excess body weight capable of breeding immediately upon emergence (Brown, 1967; Falk & Millar, 1987). Cold spring weather conditions can extend the emergence date thereby shortening the active interval. Offspring born later in the season may have relatively lower survival rates due to low body mass, while females that have late litters are likely to have reduced survival due to delayed hibernation and associated energetic costs. Consequently, montane Z. h. luteus probably only have a single litter each season.

The situation at BANWR is more complicated. This population of Z. h. luteus has an active interval equivalent to eastern US subspecies and it is slightly longer than for Z. h. preblei. In addition, it has a reproductive phenology similar to Z. h. preblei (Meaney et al., 2003). In both Z. h. preblei and eastern subspecies of Z. hudsonius there are two peaks in reproduction (Meaney et al., 2003). In eastern subspecies, females can give birth to two (possibly three) litters per year (Nichols & Conley, 1982). Given similarity of the active season between Z. h. preblei and eastern US Z. hudsonius, it is commonly believed that Z. h. preblei also can produce two, and possible three, litters per year (e.g., Armstrong, Fitzgerald & Meaney, 2011). However, I am not aware of any published reports confirming more than one litter per year in Z. h. preblei. Likewise, I found no evidence for more than one litter per year in Z. h. luteus, although it seems conceivable given similarity in length of the active season compared to eastern US subspecies. Regardless, with two peaks in reproduction, populations of Z. hudsonius are expected to expand following the second peak due to a pulse of young of the year animals entering the population (Nichols & Conley, 1982; Meaney et al., 2003). However, field studies on Z. h. luteus conducted during the last two decades found no detectable expansion of the population during summer, though there is a small pulse of presumably late litter young at the end of the active season (Table 3). Though sample sizes are small, this suggests that there may be little regular production of the crucial early litters at BANWR.

Some possible explanations for extrinsic factors that could reduce reproduction in the BANWR population include lethal genetic abnormalities due to inbreeding depression or reduced opportunity to find mates in this exceptionally small population. However, a more compelling possibility is low body weight of females emerging from hibernation. Periodic arousal from hibernation, which can be caused by warm ambient temperatures, is energetically costly and can deplete fat reserves (French & Forand, 2000). Thus, unusually warm or variable temperatures could result in reduced overwinter survival and excessively low body weights in jumping mice upon emergence (Schorr, Lukacs & Florant, 2009). Ultimately, low body weight in adult females results in lowered reproductive outputs for a population and lowered survival for offspring. Consequently, climate changes that shift phenologies earlier in the spring or that result in increasing depth or duration of warm spells during the winter may cause reduced survival and reproduction in Z. h. luteus, especially populations occurring at locations with warmer temperature equivalents. Earlier emergence times could leave metabolically stressed animals in spring without food sources (if the important spring food plants do not also shift to earlier phonologies), while winter warm spells increase potential for arousals that pose metabolic challenge for overwinter survival and subsequent reduced reproduction.

For the population of Z. h. luteus at BANWR, several factors may account for paucity of detectable above ground activity during August and September. First, adult females might be in nests with nursing young during August (Fig. 4). However, if a large proportion of females were nesting in August we would expect a pulse of young of the year in the trappable population in September. That this happens is suggested by the older specimen data, but not for the more recent field studies (Table 3). Data from Najera (1994) revealed that relative abundance (captures per 100 trap-nights) was highest in May (0.76), which tapered off in June (0.32) and July (0.24), and a second minor peak in September (0.08) and October (0.33). In the study by Wright (2012) there was no obvious second peak; relative abundance varied from 0.33 during 12–29 May, 0.42 during 11 June–8 July, 0.27 during 20 July–17 August, 0.02 during 23 August–30 September and 0.05 during 1–25 October. Second, as in the edible dormouse (Glis glis), it is possible that some females die due to depletion of energy reserves during reproduction (Bieber et al., 2014). However, this would not account for lack of detectability of other demographic groups. Third, as in the western harvest mouse (Reithrodontomys megalotis), it is possible that animals change behavior during mid-summer to become more scansorial within the herbaceous vegetation (i.e., “herbeal” sensu Wright & Frey, 2014), which might reduce capture rates in traps set on the ground (Cummins & Slade, 2007). However, based on observations made during a radio-telemetry study, Z. h. luteus appeared highly scansorial while foraging throughout the active season (Wright & Frey, 2014). Thus, there is no reason to presume capture rates would change based on a seasonal change in locomotion.

A fourth possible explanation for the paucity of detectable above ground activity during August and September is that adult males and non-reproductive adult females might enter a prolonged torpor during summer, which may or may not be confluent with winter hibernation. Prolonged torpor during hot periods is called estivation, although physiologically it is similar to hibernation during a cold period (Davis, 1976; Geiser, 2010). In some species annual hibernation routinely begins as early as August within some demographic groups (e.g., Dobson, Badry & Geddes, 1992; Sheriff et al., 2011). In other hibernating species, a hiatus of aboveground activity or summer torpor may occur as a result of drought, failure to reproduce, or failure of key food resources (Munroe, Thomas & Humphries, 2008; Bieber & Ruf, 2009). Ecologically, torpor also is beneficial to hibernating small mammals because it usually confers higher survival rates due to reduced predation (Bieber & Ruf, 2009; Turbill, Bieber & Ruf, 2011; Bieber et al., 2014). Thus, a longer growing season at BANWR that permits earlier reproduction relative to other populations of Z. h. luteus, might allow adults the luxury to enter hibernation earlier than other populations. However, there is a tradeoff between lower mortality rates during hibernation and a reduced opportunity for breeding. Thus, such a strategy may prove maladaptive if there is a decline in production of early litters and animals hibernate early rather than produce a late litter. Given similarity of climate, it is possible that other populations in the middle Rio Grande and Verde River exhibit the same pattern. Similarly, summer torpor may explain why Z. h. luteus has become undetectable at certain locations during excessively dry conditions (e.g., Frey, 2013). Behavioral changes that reduce capture probabilities, such as animals becoming more scansorial or estivating, can be misinterpreted as temporary emigration (Cummins & Slade, 2007). Thus, high rates of temporary emigration reported for Z. h. preblei (Meaney et al., 2003) may need to be reconsidered in light of possible estivation or other behavioral changes.

Management Implications

It is unknown to what extent the variation observed in phenology of hibernation and reproduction in populations of Z. h. luteus is a product of selection or phenotypic plasticity. Evolutionary control of hibernation is indicated by similarity of patterns in closely related species (Ruf & Geiser, 2014). However, studies on other species of small mammals have shown that populations living in nearby but different environments are able to use phenotypic plasticity to adjust phenology in largely predictable ways (e.g., Dobson, Badry & Geddes, 1992; Lehmer et al., 2006; Sheriff et al., 2011). Thus, it is hoped that species can adapted to changes in environmental phenology due to a changing climate through phenotypic plastic or microevolution (Boutin & Lane, 2013). Ultimately, resilience of Z. h. luteus to environmental change may be linked to how well it can adjust timing and duration of life history events. However, a recent review of climate-mediated changes in the phenology of mammals found limited evidence for contemporary microevolution and relatively many instances of non-adaptive plastic responses (Boutin & Lane, 2013). Mis-matches in the phenology of an animal with its food plants or other key resources can cause non-adaptive changes that can result in reduced survival and reproduction. Any such changes could pose a significant threat to populations of Z. h. luteus, which are already small and isolated. Thus, more research is needed on timing of life history events and reproduction in Z. h. luteus including within each sex and among populations that represent a range of latitudes and elevations. Special attention should be paid to linkages between the phenology of food plants with reproductive success and linkages between overwinter survival and body condition with subsequent reproduction.

Compliance surveys, which have the objective of determining presence or absence of Z. h. luteus at a project site must occur during times when animals are reliably detectable, if present. I recommend that such surveys occur after all overwintering adults have emerged from hibernation, which may take several weeks following first emergence. In addition, surveys conducted in late summer or fall should consider that adults might enter hibernation in August and that only late litter young of the year are active into fall. Thus, for most populations except in the middle Rio Grande valley, I recommend that compliance surveys are best implemented in July and August. In some cases it might be justifiable to survey montane locations during the last week in June or first two weeks of September, but such surveys might offset lower abundances with a larger sampling effort (i.e., larger number of trap-nights) over more nights. In the middle Rio Grande valley, and possibly other locations with a relatively high temperature equivalent (Table 2), I recommend that compliance surveys are best implemented during the last week in May through July. However, during years when winter weather occurs later than usual in the spring which may result in delayed emergence of Z. h. luteus, the start date for surveys also should be delayed.

Supplemental Information

Table S1 Specimen data used in the evaluation of the phenology

Specimen data used in the evaluation of the phenology of hibernation and reproduction in the New Mexico meadow jumping mouse (Zapus hudsonius luteus). Museums acronyms are defined in text. Julian date is the date of capture. Age class is described in text.

Click here for additional data file.

Table S2 Reproductive data for pregnant jumping mice at BANWR

Reproductive data for female meadow jumping mice (Zapus hudsonius luteus) that were field-evaluated to be pregnant at Bosque del Apache National Wildlife Refuge, Socorro County, New Mexico, 2009–2010 (Wright, 2012; Frey & Wright, 2012).

Click here for additional data file.

I am grateful to Greg Wright for his assistance and the many insightful discussions we have had about jumping mice. I thank Scott Wait of Colorado Parks and Wildlife and Jennifer L. Zahratka for providing information about jumping mice at Sambrito Creek. I thank Christina Kenny for assistance creating the histograms. I thank FS Dobson, an anonymous reviewer, and the academic editor, D Kramer, for constructive suggestions that greatly improved the paper.

Additional Information and Declarations

Competing Interests

Author Contributions

The author declares there are no competing interests.

Jennifer K. Frey conceived and designed the experiments, performed the experiments, analyzed the data, contributed reagents/materials/analysis tools, wrote the paper, prepared figures and/or tables, reviewed drafts of the paper.

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
