# Peer review of "Variation in phenology of hibernation and reproduction in the endangered New Mexico meadow jumping mouse (Zapus hudsonius luteus)"

_PeerJ, doi:10.7717/peerj.1138_

## Round 0.1 · original submission · Minor Revisions

Overview: This manuscript examines the phenology of hibernation and reproduction in the luteus subspecies of the meadow jumping mouse in relation to altitude and latitude and compares it to the phenology of related species and subspecies. The original data are extracted from museum specimens to supplement a limited number of previous field studies. Although not stated explicitly, I infer that this subspecies is rarely captured, necessitating use of museum specimens and conclusions based on limited sample sizes, incomplete information, and practically no statistical analysis. The author concludes that montane populations emerge from hibernation later than valley populations, and there is a clear negative relationship between the average temperature expected from altitude and latitude and the date of emergence. In contrast, she concludes that populations in both areas immerge to their burrows at about the same time. Thus, the active interval is about 30 d shorter in the montane environment. Reproduction is apparently somewhat later in montane than in valley populations. A period of limited aboveground activity in late summer was documented in one low altitude, southern population.

Reviews: The reviewers find that the manuscript makes a useful contribution, but that it needs to be placed in a broader context, including work on other species. They both suggest some useful references as a starting point. I concur with this view. It is not necessary to provide a complete review of the literature on altitudinal and latitudinal patterns in hibernation and reproduction. However, you should indicate broadly the current understanding of these research questions.

Another aspect of placing your study in context is the need to clarify why it is needed. The amount of literature reviewed could suggest that the patterns are quite well understood, so you need point out the gaps to make it clear to readers what is missing and how your study will help to fill those gaps. You should also clarify whether there is an issue of limited data or other reasons that someone would seek out museum specimens lacking important information about sampling effort to draw ecological inferences.

I also strongly agree with reviewer 1 that the compilation of observations in your Results/Discussion section needs a stronger synthesis (i.e. a true discussion, not just results). At the end of each section, we need an overall summary statement of your conclusions and the strength of support for them, somewhat like you did for reproduction in BANWR. For example, if you conclude that immergence times are similar in montane and valley environments, this should be explicit at the end of the appropriate Results/Discussion session along with an indication of how well the data support this conclusion. The last point is important because I think you could be a bit more critical in evaluating the data. In some places, you refer to limited sample sizes or trapping effort during periods without captures, but you do not seem to consistently incorporate such insights into your conclusions. Given the small sample sizes and diverse origins of the data, I can understand the lack of statistical analysis, but it does reduce the validity of the conclusions; a more critical approach to the reliability of the conclusions would go some way to compensating for the lack of formal statistics. A discussion normally includes placing your findings in the context of previous research. As you have organized your manuscript, this might fit better in what you call conclusions. But you should make an effort to synthesize what new insights you have provided with regard to your questions and whether any of your findings have implications for the broader field.

The reviewers have made a number of other helpful selections, which you should consider carefully. I also have some reviewer-level suggestions for the manuscript, based partly on my own experience with hibernation and activity in eastern chipmunks.

Editor’s comments:
The decision not to include line numbers makes it difficult to refer to specific points in the text. When specific wording is referred to, my comments are indicated by page/paragraph/line, with page referring to numbering on the actual page, not the pdf page number. In a revised version, line numbers would be very helpful.

Abstract
• 1/1/8 "similar to other populations" is confusing since you have just been comparing montane and valley populations. I think this refers to the eastern subspecies, not just other populations of the same subspecies?
• You state that a goal was to examine timing of reproduction but refer only to number of litters. A clearer statement on timing is needed.
• The final, rather vague sentence could be replaced by a summary statement that would include a brief justification for the recommendation.
Introduction
• 2/1/3. Do you think that a 40-yr old reference is still appropriate for a comparative statement on the duration of hibernation? You can probably make a more specific statement by looking at recent reviews and/or considering recent studies of long-duration hibernators such as dormice.
• If the Introduction explicitly identified the other species and sub-species of jumping mice that have been studied and their approximate ranges, later comparisons would be much easier to follow.
Methods
• 5/2/1ff. Readers should be able to determine more precisely the locations sampled. A map would be helpful. However, Table 2 provides much of the required information, needing only the addition of longitude and the classification category (montane/valley).
• 5/2/11ff. Wouldn't it be easier for readers if you converted your equation to SI units directly? If you want readers to have access to the Imperial unit equation, you could provide both. At any rate, you must clarify the equation by providing unambiguous units for temperature equivalent and making certain that minus signs do not appear to be hyphens. Note that the abscissa in Fig. 2 also lacks units.
• 5/2/1,4. Reword. 'Data' is plural.
Results/Discussion, Tables, Figures
• 7/1/10. I think you mean 'imminently' not 'eminently'.
• 7/1/13. This does not seem to be the place to refer to arousal or provide a dated reference to its possible justification. There is a lot of research on this topic. It occurs in all hibernators, as far as I am aware and the literature on costs and benefits of arousal is very large and growing. If you are trying to make a point about how arousal would affect interpretation of winter trapping, say so explicitly.
• 7/1/17. 'trap-nights during warm days' is not clear: daytime trapping or night trapping during warm periods?
• 7/1/19. Lack of trapping success is not sufficient evidence for a strong conclusion that a species hibernates. Without body temperature data, you can only suggest that they probably hibernate. All you really know is that they were not likely to have been aboveground and foraging.
• 7/3/1ff. The sub-section on 'Problems Interpreting Reproduction during Field Studies' seems more appropriate for Methods. The title is also a bit awkward. It seems to me that you are referring to problems identifying reproductive state.
• 8/2/11. Not clear what you mean by reference to a burrow that contained fetuses. Perhaps you meant to refer to the female?
• 10/3/1ff. Regarding a decrease in aboveground activity in late summer, it is worth noting that there is a fairly substantial literature on the so-called 'summer lull' in eastern chipmunks, although a well supported functional explanation still eludes researchers. Eastern chipmunks often greatly reduce aboveground activity for a period in August, sometimes not re-emerging until the following spring if there is no mast in the fall (Munro et al 2008, Can J Zool and Dunford and Yahner cited therein, among others).
• Fig. 2. Horizontal lines not needed. Precision of slope should be same as intercept. Units are needed on the x-axis.
• Fig. 3. Horizontal lines not needed.
• Fig. 4. Caption is a bit awkward. Documented by evidence vs. inferred from other evidence. Clarify what evidence provides documentation and what evidence allows inference.
Acknowledgements
• 13/2/4. anonymous (sp)

Reviewer 1 ·

Basic reporting

Please see comments below.

Experimental design

Please see comments below under Methods.

Validity of the findings

Please see comments below.

Additional comments

Review. In this ms Frey describes the hibernation and breeding patterns of jumping mice living in different environments – valley populations and montane populations. She finds that those mice living in valley populations have earlier spring phenology but similar entrance into hibernation as mice from montane populations. Although this is a very detailed account of jumping mice I found that it lacks broad applicability as written. I believe Frey could greatly improved this ms by placing her findings into a broader context of hibernation (and breeding) phenology relative to environmental constraints. There has been considerable work done on other similar (long duration of hibernation) hibernating species which has been overlooked. I suggest a major revision.
I suggest a review and inclusion of the following papers as a starting point for the revision.
Buck et al. 2008. In: Lovegrove BG, McKechnie AE (eds) Hypometabolism in animals: hibernation, torpor, and cryobiology. Proceedings of the 13th international hibernation symposium. Pietermaritzburg, pp 317–326
Florant and Healy 2011. J Comp Physiol B doi:10.1007/s00360-011-0630-y
Geiser 2004 Annu Rev Physiol 66: 239-274
Inouye et al. 2001 Proc Natl Acad Sci 97:1630-1633
Ozgul et al. 2010 Nature
Sheriff et al. 2011 Proc R Soc B
Sheriff et al. 2013 Phil Trans R Soc B

Introduction.
In light of my comments above I suggest that the first paragraph broadly cover the ‘problem’ at hand – phenological differences in hibernators (of the same species) experiencing different environments. The introduction should cover details of this problem and introduce information pertaining to the predictions and not focus solely on jumping mice. I would suggest only a single paragraph detailing jumping mice. These changes would not alter the focus of the ms just place it into a broader context.
As a minor comment I suggest deleting the term ‘profound hibernator’ in the first sentence of the introduction. I think that this distinction is becoming archaic. Williams et al. 2011 J Comp Physiol. has a nice definition of a hibernator which encompasses all hibernators.
Methods.
I think there is much information missing from the methods that is needed to allow an appropriate assessment of the validity of the study.
1) Frey makes reference to using ‘information on field studies from published literature and unpublished reports’ at the end of the first paragraph in Methods. Please specify (reference) these sources. Possibly in a supplemental table?
2) I’m very unsure of how emergence and immergence dates were calculated. Please explain further how histograms of specimens by julian date (I'm assuming collection times?) is related to emergence or entrance? This is obviously the crux of the methodology and needs very clear and precise information.
3) Last line of the methods section. At age class 0.35 their second active season please clarify that this is their first breeding season and thus they are adults at this point? Further on this point why were specimens assigned to 1-6 or 1-8 age class groups but then the results and discussion only refer to adults and young-of-the-year. If it is only relevant to split the age classes into YOY and adult then only include this information.
Results and Discussion.
Throughout the results/discussion I found it simply a list of timing results of a given sex, site, population etc. but very little discussion or comparison. I think this section could greatly benefit from a detailed discussion/comparison of the general differences found between site type (valley vs. montane), sex, age (etc. as appropriate) and the ecological importance/implications of these differences and of potential factors leading to these differences.
Conclusion.
Similarly to the R/D section this lacked a general, broad ecological conclusion.

·

Basic reporting

See attached marked copy of the manuscript.

Experimental design

There is no problem with the study design.

Validity of the findings

The results are appropriately interpreted.

Additional comments

This study examined activity and reproductive patterns of meadow jumping mice in western North America. Quite a bit of information is brought together, reviewed, and summarized. I chose to review the manuscript because the activity cycles of the mice are fascinating and little known. I expected to learn interesting details, and I was not disappointed.

The study is largely descriptive, though predictions are suggested. These “predictions” are more like expectations based on knowledge of the natural history of the species and subspecies, and are reasonable. However, the discussion does not treat them one by one (i.e., does not restate them overtly), although all the information is there. The study is also not placed in a wider context, as is available from studies of other hibernators. I’m most aware of my own work, of course, but a quick Web of Science search might turn up additional work that would place jumping mice in a larger context. For example, Dobson and Davis (1986) reviewed activity and hibernation timing in California ground squirrels, showing a wide range of activity patterns over geography. Dobson et al. (1992, Can J Zool) compared patterns of activity of Columbian ground squirrels at high and low elevations, and showed strong differences in a species that has an even shorter activity period than the jumping mice (though of course the ground squirrels are much larger). Both studies provide a rationale to expect significant variation among environmental conditions in the activity periods of jumping mice.

I’ve made some comments on a copy of the text, and I hope that some of them are helpful. This study is very well written and very informative. And very carefully done. Thanks very much for teaching me so much about these fascinating small mammals. If I can help further, please contact me directly.


F. Stephen Dobson
fsdobson@msn.com

---

## Round 0.2 · Minor Revisions

Your manuscript is greatly improved and I am prepared to accept it as soon as a few clarifications and corrections of minor typos have been made. I am pleased that you found the suggestions by the reviewers and by me helpful.

L23-25 The contrast between reproduction and above-ground activity will make little sense to readers of the Abstract only. I suggest explaining little or no evidence of second reproduction with combined explanations of later first reproduction and possible period of late summer inactivity.
L53. have, not has
L72. You might modify this sentence so that your intended point is clearer because radio collars allow animals to be detected in hibernation.
L102. phenologies (sp)
L150. dependent (sp)
L440-442. This part of the summary appears to relate to the hibernation rather than to the reproduction sub-section.
L532. litter (sp)
L593. Do you mean 'arboreal' rather than 'herbeal'? (I don't know whether the word 'herbeal' even exists. Perhaps this is a case of over-correction by Word?)
L649. What do you mean by 'larger trap-nights'?
Table 1 meadow jumping (add space)
Table 3 by month (add space)

---

## Author Rebuttal · Round 0.2

*COLLEGE OF AGRICULTURAL, CONSUMER AND ENVIRONMENTAL SCIENCE*

*Department of Fish, Wildlife and Conservation Ecology*
*MSC 4901*
*New Mexico State University*
*P.O. Box 30003*
*Las Cruces, NM 88003-8003*
*Phone: (575) 646-1544*
*Fax: (575) 646-1281*
*E-mail: FWCE@nmsu.edu*

[Figure]

30 June 2015

Dear Editor,

Herewith, please find a revised version of a manuscript originally titled "Timing of hibernation and reproduction in the endangered New Mexico meadow jumping mouse (*Zapus hudsonius luteus*)", which is now titled "Variation in the phenology of hibernation and reproduction in the endangered New Mexico meadow jumping mouse (*Zapus hudsonius luteus*)". I appreciate the time spent on this manuscript by you and the reviewers and I am grateful for the helpful comments and suggestions. On basis of the reviews, I have made a major revision to the manuscript and I feel it is greatly improved. Importantly, I expanded the context of the paper from the narrow perspective of the study organism, to the broader field of research on hibernation phenology in other species. This process gave me important new perspectives on various issues related to *Z. h. luteus*, which are likely to have important impact on its conservation and management. Below (in red) I have detailed how I handled each of the comments and suggestions made by you and the reviewers. In addition to those, I also made a many changes to the manuscript to improve its organization and clarity. This also included a reanalysis of the predicted dates of first emergence based on an improved regression equation. I am grateful for being given the opportunity to revise this manuscript and I look forward to your feedback.

Sincerely,

Jennifer K. Frey

College Associate Professor
Research Area: Ecology and Conservation of Mammals

# Response to Editor and Reviewers

## Editor's comments

*Overview: This manuscript examines the phenology of hibernation and reproduction in the luteus subspecies of the meadow jumping mouse in relation to altitude and latitude and compares it to the phenology of related species and subspecies. The original data are extracted from museum specimens to supplement a limited number of previous field studies. Although not stated explicitly, I infer that this subspecies is rarely captured, necessitating use of museum specimens and conclusions based on limited sample sizes, incomplete information, and practically no statistical analysis.*

I revised the first paragraph of the Methods section (line 200-204) to explicitly state the reason for using museum specimens as the primary source of data. I also included caveats to data limitations throughout the manuscript.

*The author concludes that montane populations emerge from hibernation later than valley populations, and there is a clear negative relationship between the average temperature expected from altitude and latitude and the date of emergence. In contrast, she concludes that populations in both areas immerge to their burrows at about the same time. Thus, the active interval is about 30 d shorter in the montane environment. Reproduction is apparently somewhat later in montane than in valley populations. A period of limited aboveground activity in late summer was documented in one low altitude, southern population.*

*Reviews: The reviewers find that the manuscript makes a useful contribution, but that it needs to be placed in a broader context, including work on other species. They both suggest some useful references as a starting point. I concur with this view. It is not necessary to provide a complete review of the literature on altitudinal and latitudinal patterns in hibernation and reproduction. However, you should indicate broadly the current understanding of these research questions.*

I reframed the paper to move it from the narrow context of jumping mice, to the broader body of research on the phenology of hibernation in general. This included the addition of many new references. In particular, I added two new paragraphs to the beginning of the introduction (lines 30-70). Other references and broader context are also woven through other sections of the paper. I believe this greatly improved the paper and I thank the editor and reviewers for their suggestions.

*Another aspect of placing your study in context is the need to clarify why it is needed. The amount of literature reviewed could suggest that the patterns are quite well understood, so you need point out the gaps to make it clear to readers what is missing and how your study will help to fill those gaps.*

I made a major revision to the introduction. It now includes several new paragraphs that explain the context and relevance of my results (see especially lines 71-106). In particular, please note the new third paragraph (lines 71-85). It provides an explicit explanation for why data on phenology is necessary for Z. h. luteus. In essence, this taxon has been listed as an endangered species, which requires surveys and monitoring as part of its management. It is not possible to conduct these kinds of studies without understanding its annual cycle since jumping mice cannot be detected while hibernating. The fourth paragraph (lines 86-106) goes on to summarize what little is known about the phenology of other jumping mice and that it cannot be extrapolated to Z. h. luteus.

*You should also clarify whether there is an issue of limited data or other reasons that someone would seek out museum specimens lacking important information about sampling effort to draw ecological inferences.*

*I revised the methods to explain why extracting data from museum specimens was necessary (line 200-206). In essence, Z. h. luteus has declined to a very few populations, most of which are exceptionally small. Intensive study of these small populations is not feasible because it would require invasive methods including trapping, which can pose a risk. Most of the wild populations are simply not good candidates for field research.*

*I also strongly agree with reviewer 1 that the compilation of observations in your Results/Discussion section needs a stronger synthesis (i.e. a true discussion, not just results). At the end of each section, we need an overall summary statement of your conclusions and the strength of support for them, somewhat like you did for reproduction in BANWR. For example, if you conclude that immergence times are similar in montane and valley environments, this should be explicit at the end of the appropriate Results/Discussion session along with an indication of how well the data support this conclusion. The last point is important because I think you could be a bit more critical in evaluating the data. In some places, you refer to limited sample sizes or trapping effort during periods without captures, but you do not seem to consistently incorporate such insights into your conclusions. Given the small sample sizes and diverse origins of the data, I can understand the lack of statistical analysis, but it does reduce the validity of the conclusions; a more critical approach to the reliability of the conclusions would go some way to compensating for the lack of formal statistics. A discussion normally includes placing your findings in the context of previous research. As you have organized your manuscript, this might fit better in what you call conclusions. But you should make an effort to synthesize what new insights you have provided with regard to your questions and whether any of your findings have implications for the broader field.*

*I reorganized the manuscript so that the Results and Discussion are separate sections (rather than a combined Results/Discussion and a Conclusions). This necessitated reorganizing some material to fit the new structure. For instance, the discussion on the possible reasons for the mid-summer hiatus in above ground activity at BANWR was moved from Results to Discussion (e.g., lines 578-619). Within the Results, I added a summary for each subsection (see lines 336-346, 376-389). In addition, throughout the Results and Discussion I endeavored to explicitly state data limitations and scope of inference. I believe that these changes served to greatly improve the paper.*

*The reviewers have made a number of other helpful selections, which you should consider carefully. I also have some reviewer-level suggestions for the manuscript, based partly on my own experience with hibernation and activity in eastern chipmunks.*

**Editor's comments:**
*The decision not to include line numbers makes it difficult to refer to specific points in the text. When specific wording is referred to, my comments are indicated by page/paragraph/line, with page referring to numbering on the actual page, not the pdf page number. In a revised version, line numbers would be very helpful.*
*I apologize for not including line numbers in the original version. I have included line numbers in the revision.*

**Abstract**
*• 1/1/8 "similar to other populations" is confusing since you have just been comparing*

*montane and valley populations. I think this refers to the eastern subspecies, not just other populations of the same subspecies?*
Corrected (line 16). The comparison was with other subspecies, including the western subspecies Z. h. preblei.

*• You state that a goal was to examine timing of reproduction but refer only to number of litters. A clearer statement on timing is needed.*

Corrected (line 21-23). I added time frames for pregnant females for montane populations and BANWR.

*• The final, rather vague sentence could be replaced by a summary statement that would include a brief justification for the recommendation.*
Corrected (line 26-27) . I attempted to briefly summarize and justify the recommendation.

**Introduction**
*• 2/1/3. Do you think that a 40-yr old reference is still appropriate for a comparative statement on the duration of hibernation? You can probably make a more specific statement by looking at recent reviews and/or considering recent studies of long-duration hibernators such as dormice.*

I made a major revision to the introduction (lines 31-198). As a consequence, this sentence was deleted. The new introduction includes general information about mammalian hibernators to put this study into perspective.

*• If the Introduction explicitly identified the other species and sub-species of jumping mice that have been studied and their approximate ranges, later comparisons would be much easier to follow.*
In the major revision of the introduction, I added a new paragraph that describes each species of jumping mouse, including their distribution (line 108-126)

**Methods**
*• 5/2/1ff. Readers should be able to determine more precisely the locations sampled. A map would be helpful. However, Table 2 provides much of the required information, needing only the addition of longitude and the classification category (montane/valley).*
Corrected. I added longitude to Table 2; montane and valley populations are defined in the methods (line 221-223). Because of the topographic complexity of the region, I feel the table conveys information about pertinent geographic variation better than a map. The table also allows inclusion of other relevant data such as temperature equivalent and predicted date of immergence.

*• 5/2/11ff. Wouldn't it be easier for readers if you converted your equation to SI units directly? If you want readers to have access to the Imperial unit equation, you could provide both. At any rate, you must clarify the equation by providing unambiguous units for temperature equivalent and making certain that minus signs do not appear to be hyphens. Note that the abscissa in Fig. 2 also lacks units.*
Corrected (line 236-237). I changed the equation and all results/figure to metric.

*• 5/2/1,4. Reword. 'Data' is plural.*
Corrected throughout.

*Results/Discussion, Tables, Figures*

• *7/1/10. I think you mean 'imminently' not 'eminently'.*
Corrected throughout.

• *7/1/13. This does not seem to be the place to refer to arousal or provide a dated reference to its possible justification. There is a lot of research on this topic. It occurs in all hibernators, as far as I am aware and the literature on costs and benefits of arousal is very large and growing. If you are trying to make a point about how arousal would affect interpretation of winter trapping, say so explicitly.*

Corrected.  I deleted the sentence referring to arousal. The main point was to counter the speculation that Z. h. luteus might not hibernate at BANWR.  This is now stated in line 382-387.

• *7/1/17. 'trap-nights during warm days' is not clear: daytime trapping or night trapping during warm periods?*
Corrected (lines 361-364).  Clarified by changing "days" to "spells"

• *7/1/19. Lack of trapping success is not sufficient evidence for a strong conclusion that a species hibernates. Without body temperature data, you can only suggest that they probably hibernate. All you really know is that they were not likely to have been aboveground and foraging.*
Corrected and clarified (line 382-386).

• *7/3/1ff. The sub-section on 'Problems Interpreting Reproduction during Field Studies' seems more appropriate for Methods. The title is also a bit awkward. It seems to me that you are referring to problems identifying reproductive state.*
Corrected.  I moved this section to the methods and changed the subtitle (line 265-295).  In addition, I rearranged the methods to contain all information regarding reproductive state of museum species in its own paragraph separate from information from field studies (line 255-264).

• *8/2/11. Not clear what you mean by reference to a burrow that contained fetuses. Perhaps you meant to refer to the female?*
Corrected.  I revised the sentence to clarify (line 287-288).

• *10/3/1ff. Regarding a decrease in aboveground activity in late summer, it is worth noting that there is a fairly substantial literature on the so-called 'summer lull' in eastern chipmunks, although a well supported functional explanation still eludes researchers. Eastern chipmunks often greatly reduce aboveground activity for a period in August, sometimes not re-emerging until the following spring if there is no mast in the fall (Munro et al 2008, Can J Zool and Dunford and Yahner cited therein, among others).*

I moved this section to the discussion and totally revamped it to bring in more information from other species (see lines 598-620).  It helped to reinforce some ideas that I had not explicitly included in the original version.  I now explicitly discuss the possibility of aestivation and/or early immergence into hibernation as possible reasons for the reduction in above ground activity.  It also helps to explain some instances where surveys have failed to detect any activity during some years with unusually dry conditions.  I think this section is much improved and appreciate the comments.

*• Fig. 2. Horizontal lines not needed. Precision of slope should be same as intercept. Units are needed on the x-axis.*
*Corrected.This now figure 3.*

*• Fig. 3. Horizontal lines not needed.*
*Corrected.This is now figure 1*

*• Fig. 4. Caption is a bit awkward. Documented by evidence vs. inferred from other evidence. Clarify what evidence provides documentation and what evidence allows inference.*
*Corrected. I clarified the definitions (see lines 864-869)*

*Acknowledgements*
*• 13/2/4. anonymous (sp)*
*Corrected.*

## Reviewer Comments

## Reviewer 1 (Anonymous)

### Basic reporting

*Please see comments below.*

### Experimental design

*Please see comments below under Methods.*

### Validity of the findings

*Please see comments below.*

### Comments for the author
*Review. In this ms Frey describes the hibernation and breeding patterns of jumping mice living in different environments – valley populations and montane populations. She finds that those mice living in valley populations have earlier spring phenology but similar entrance into hibernation as mice from montane populations. Although this is a very detailed account of jumping mice I found that it lacks broad applicability as written. I believe Frey could greatly improved this ms by placing her findings into a broader context of hibernation (and breeding) phenology relative to environmental constraints. There has been considerable work done on other similar (long duration of hibernation) hibernating species which has been overlooked. I suggest a major revision.*
*I suggest a review and inclusion of the following papers as a starting point for the revision. Buck et al. 2008. In: Lovegrove BG, McKechnie AE (eds) Hypometabolism in animals: hibernation, torpor, and cryobiology. Proceedings of the 13th international hibernation symposium. Pietermaritzburg, pp 317–326*
*Florant and Healy 2011. J Comp Physiol B doi:10.1007/s00360-011-0630-y*
*Geiser 2004 Annu Rev Physiol 66: 239-274*

*Inouye et al. 2001 Proc Natl Acad Sci 97:1630-1633*
*Ozgul et al. 2010 Nature*
*Sheriff et al. 2011 Proc R Soc B*
*Sheriff et al. 2013 Phil Trans R Soc B*

*Thank you for your very helpful comments. In preparing my revision, I read widely within the literature on the phenology of hibernation, including the papers you suggested. I have reframed the paper to put it in context of other research. In so doing, I added these and many other new references. I gained a great deal of new insight during this process and I think it served to dramatically improve the paper. See especially the two new introductory paragraphs (lines 30-70).*

**Introduction.**

*In light of my comments above I suggest that the first paragraph broadly cover the 'problem' at hand – phenological differences in hibernators (of the same species) experiencing different environments. The introduction should cover details of this problem and introduce information pertaining to the predictions and not focus solely on jumping mice. I would suggest only a single paragraph detailing jumping mice. These changes would not alter the focus of the ms just place it into a broader context.*

*I made a major revision to the paper, including an overhaul of the introduction. I broadened the context of the paper by adding information on mammalian hibernators, in general, to the introduction. I believe these changes have greatly improved the paper. See especially the two new introductory paragraphs (lines 30-70). I did keep information on jumping mice, but in a separate section of the introduction (lines 107-189). No other paper has provided a review of information on phenology of hibernation and reproduction in jumping mice. I think this was necessary in order to generate the specific expectations and to provide a context for comparison.*

*As a minor comment I suggest deleting the term 'profound hibernator' in the first sentence of the introduction. I think that this distinction is becoming archaic. Williams et al. 2011 J Comp Physiol. has a nice definition of a hibernator which encompasses all hibernators.*

*I deleted the term profound hibernator. In the introduction, I added a definition to hibernation that makes it clear exactly what I am referring to (line 33-35). I cited a different paper for the definition, but I did cite the Williams et al paper in a different place within the manuscript.*

**Methods.**

*I think there is much information missing from the methods that is needed to allow an appropriate assessment of the validity of the study.*

*1) Frey makes reference to using 'information on field studies from published literature and unpublished reports' at the end of the first paragraph in Methods. Please specify (reference) these sources. Possibly in a supplemental table?*

*I added the references to the sentence and separated them by type (published literature, theses, agency reports). See lines 203-206.*

*2) I'm very unsure of how emergence and immergence dates were calculated. Please explain further how histograms of specimens by julian date (I'm assuming collection times?) is related to emergence or entrance? This is obviously the crux of the methodology and needs very clear and precise information.*

*I added information to clearly identify what the Julian date signifies (day of capture) and that emergence and immergence dates based on these data may be imperfect. See lines 223-227.*

*3) Last line of the methods section. At age class 0.35 their second active season please clarify that this is their first breeding season and thus they are adults at this point? Further on*

*this point why were specimens assigned to 1-6 or 1-8 age class groups but then the results and discussion only refer to adults and young-of-the-year. If it is only relevant to split the age classes into YOY and adult then only include this information.*

*There are no field studies that have followed multiple individuals through time in order to generate age specific information. For the museum specimens, the only way to obtain relative age data was by examining cranial and dental characteristics. However, no study has linked the morphological "age classes" based on cranial and dental characteristics to known age individuals. Thus, the age classes present a continuum of ages, but exact age is mostly unknown. The exception is that animals that have emerged from their first hibernation appear to be at age class 0.35. Thus, we can distinguish between young of the year (age class < 0.35) and "adults" that have gone through at least one hibernation (age class ≥ 0.35). This is easily seen in Figure 1, which is a scatterplot of age class by Julian date wherein all individuals emerging from hibernation have age class > 0.35 and later in the season there is a pulse of < 0.35 juveniles. Consequently, I provided a major revision to this section and directly referenced the figure as the basis for my interpretation. See lines 245-254.*

**Results and Discussion.**
*Throughout the results/discussion I found it simply a list of timing results of a given sex, site, population etc. but very little discussion or comparison. I think this section could greatly benefit from a detailed discussion/comparison of the general differences found between site type (valley vs. montane), sex, age (etc. as appropriate) and the ecological importance/implications of these differences and of potential factors leading to these differences.*

*I made a major revision to this section. Importantly, I separated the Results/Discussion section into two separate sections. The Results section now presents the results more succinctly and includes summaries of each subsection (see lines 296-482). The new Discussion now incorporates information for other species to broaden the context of the discussion. See lines 483-620.*

**Conclusion.**
*Similarly to the R/D section this lacked a general, broad ecological conclusion.*
*I eliminated the Conclusion section from the paper. Most of this material is now in the new Discussion. The new Discussion has a broader context as it incorporates information from the broader literature on hibernation phenology. See lines 483-620.*

# Reviewer 2 (F. Stephen Dobson)

## *Basic reporting*

*See attached marked copy of the manuscript.*

## *Experimental design*

*There is no problem with the study design.*

## *Validity of the findings*

*The results are appropriately interpreted.*

## Comments for the author

*This study examined activity and reproductive patterns of meadow jumping mice in western North America. Quite a bit of information is brought together, reviewed, and summarized. I chose to review the manuscript because the activity cycles of the mice are fascinating and little known. I expected to learn interesting details, and I was not disappointed.*

*The study is largely descriptive, though predictions are suggested. These "predictions" are more like expectations based on knowledge of the natural history of the species and subspecies, and are reasonable.*

*I deleted use of the word "predictions" and referred to these as assumptions or expectation throughout the paper. For instance, see lines 190-191.*

*However, the discussion does not treat them one by one (i.e., does not restate them overtly), although all the information is there.*

*I have revamped the discussion to explicitly address each expectation. Please see paragraph 483-509. I think this served to greatly improve the discussion.*

*The study is also not placed in a wider context, as is available from studies of other hibernators. I'm most aware of my own work, of course, but a quick Web of Science search might turn up additional work that would place jumping mice in a larger context. For example, Dobson and Davis (1986) reviewed activity and hibernation timing in California ground squirrels, showing a wide range of activity patterns over geography. Dobson et al. (1992, Can J Zool) compared patterns of activity of Columbian ground squirrels at high and low elevations, and showed strong differences in a species that has an even shorter activity period than the jumping mice (though of course the ground squirrels are much larger). Both studies provide a rationale to expect significant variation among environmental conditions in the activity periods of jumping mice.*

*Thank you for your comments and suggestions. I have revamped the paper to place it within the context of our knowledge of the phenology of hibernating mammals in general. This included addition of numerous new citation, including your suggestions. I believe this has greatly improved the paper. For instance, please see the new two opening paragraphs of the Introduction, lines 30-70.*

*I've made some comments on a copy of the text, and I hope that some of them are helpful. This study is very well written and very informative. And very carefully done. Thanks very much for teaching me so much about these fascinating small mammals. If I can help further, please contact me directly.*

*All comments written on the text were incorporated into the revised manuscript.*

*F. Stephen Dobson*
*fsdobson@msn.com*

## Annotated manuscript
*The reviewer has provided feedback as annotations on the manuscript PDF.*

---

## Round 0.3 · accepted · Accept

Thank you for the quick corrections. I now find the manuscript suitable for publication.